# Whole genome sequencing reveals hidden transmission of carbapenemase-producing *Enterobacterales*

Kalisvar Marimuthu [1,2,3 ✉], Indumathi Venkatachalam[4], Vanessa Koh[1,2], Stephan Harbarth[5], Eli Perencevich[6,7], Benjamin Pei Zhi Cherng[3,4,8], Raymond Kok Choon Fong[9], Surinder Kaur Pada[10], Say Tat Ooi[11], Nares Smitasin[3,12], Koh Cheng Thoon[13], Paul Anantharajah Tambyah[12], Li Yang Hsu[3,14,15], Tse Hsien Koh[4,8], Partha Pratim De[2], Thean Yen Tan[9], Douglas Chan[10], Rama Narayana Deepak[11], Nancy Wen Sim Tee[12], Andrea Kwa[4,16,17], Yiying Cai[4,15], Yik-Ying Teo[14,15], Natascha May Thevasagayam[1,2], Sai Rama Sridatta Prakki[1,2], Weizhen Xu[1,2], Wei Xin Khong[2], David Henderson[18], Nicole Stoesser [19], David W. Eyre [20], Derrick Crook[19], Michelle Ang[1], Raymond Tzer Pin Lin [1,12], Angela Chow [2,14,21], Alex R. Cook[14], Jeanette Teo[12], Oon Tek Ng [1,2,21 ✉] & Carbapenemase-Producing Enterobacteriaceae in Singapore (CaPES) Study Group*

Carbapenemase-producing *Enterobacterales* (CPE) infection control practices are based on the paradigm that detected carriers in the hospital transmit to other patients who stay in the same ward. The role of plasmid-mediated transmission at population level remains largely unknown. In this retrospective cohort study over 4.7 years involving all multi-disciplinary public hospitals in Singapore, we analysed 779 patients who acquired CPE (1215 CPE isolates) detected by clinical or surveillance cultures. 42.0% met putative clonal transmission criteria, 44.8% met putative plasmid-mediated transmission criteria and 13.2% were unlinked. Only putative clonal transmissions associated with direct ward contact decreased in the second half of the study. Both putative clonal and plasmid-mediated transmission associated with indirect (no temporal overlap in patients' admission period) ward and hospital contact did not decrease during the study period. Indirect ward and hospital contact were identified as independent risk factors associated with clonal transmission. In conclusion, undetected CPE reservoirs continue to evade hospital infection prevention measures. New measures are needed to address plasmid-mediated transmission, which accounted for 50% of CPE dissemination.

A full list of author affiliations appears at the end of the paper.

Carbapenemase-producing *Enterobacterales* (CPE) pose a public health threat due to their rapid global dissemination as well as increased morbidity and mortality largely as a consequence of the lack of safe and efficacious treatment options in many cases[1]. In many countries, including Singapore, the incidence of CPE continues to increase despite active infection prevention efforts[2].

CPE infection prevention and control guidelines emphasize prevention of direct patient-to-patient transmission, with early detection and isolation of CPE carriers as key components of the control strategy[3]. However, multiple potential sources of CPE transmission that are not adequately addressed by current CPE infection prevention bundles have been documented. These include environmental sources[4], healthcare workers[5], hospital equipment and instruments[6], and community transmission[7]. Determining and understanding the predominant routes of CPE transmission is essential for successful control.

In addition to clonal dissemination of carbapenemase-producing organisms, carbapenemase gene transmission can occur by horizontal gene transfer via mobile genetic elements[8,9]. A pilot study suggested that plasmid conjugation may contribute to the persistence of carbapenemase genes in a hospital ecology despite aggressive infection prevention interventions[10].

To fully examine clonal CPE and plasmid-mediated transmission dynamics, we performed whole-genome sequencing on CPE isolates from a nation-wide surveillance programme encompassing six multi-disciplinary public sector hospitals accounting for 80% of inpatient hospitalized care in Singapore over a five-year period. We determined the prevalence and trend over time, as well as the epidemiologic risk factors associated with clonal and plasmid CPE transmission.

## Results

**Study population.** From September 2010 to April 2015, 1312 CPE isolates (from 817 unique patients) were submitted as part of mandatory reporting to the National Public Health Laboratory, of which 1302 (99.2%) CPE isolates were successfully cultured, whole-genome sequenced and assembled. Of the 1302 successfully assembled isolates, 1251 (96.1%) had concordant bacterial species and carbapenemase gene genomically-identified compared with laboratory phenotypic data. A further 36 isolates lacking patient or date of culture information were excluded from analysis, resulting in 1215 (93.3%) isolates analysed. A total of 779 acquisition patients were included in the final transmission analysis, of which 327 (42.0%) met criteria for putative clonal transmission and 349 (44.8%) met putative plasmid-mediated transmission criteria, while 103 (13.2%) were unlinked (Fig. 1). Considering only surveillance cultures for infection control purposes, there were 525 acquisition patients, of which 186 (35.4%) met criteria for putative clonal transmission, 231 (44.0%) met putative plasmid-mediated transmission criteria and 108 (20.6%) were unlinked. Considering only clinical cultures, there were 348 acquisition patients, of which 93 (26.7%) met criteria for putative clonal transmission, 135 (38.8%) met putative plasmid-mediated transmission criteria and 120 (34.5%) were unlinked.

The median age of acquisition patients was 68 years (interquartile range [IQR], 58–78). Of the 779 acquisition patients, 444 (57.0%) were males. The median number of admission episodes per patient was 5 (IQR, 3–9) and the median length of stay per episode was 5 days (IQR, 2–13). The majority of samples was collected as surveillance cultures ($N = 483$, 62.0%) with the rest ($N = 296$, 38.0%) collected based on clinical indications.

The bacterial species represented within the 1215 isolates was predominantly *Klebsiella pneumoniae* ($N = 532$, 43.8%), followed by *Escherichia coli* ($N = 377$, 31.0%), *Enterobacter spp* ($N = 195$, 16.0%) and *Citrobacter freundii* ($N = 66$, 5.4%). The most common species strain types were *E. coli* ST131 (63 of 377, 16.7%), *K. pneumoniae* ST14 (53 of 532, 10.0%), *Enterobacter cloacae* ST93 (35 of 195, 18.0%) and *K. pneumoniae* ST147 (34 of 532, 6.4%). The carbapenemase gene prevalence was as follows: 525 (43.2%) $bla_{KPC}$, 499 (41.1%) $bla_{NDM}$, 126 (10.4%) $bla_{OXA-type}$, 26 (2.1%) $bla_{IMP}$, 13 (1.1%) $bla_{IMI}$ and 26 (2.1%) dual-carriage. The predominant carbapenemase-encoding plasmids identified were pKPC2-9 (72947 bp; $N = 125$, 10.3%) and pNDM-ECS01 (41190 bp; $N = 98$, 8.1%).

**Putative clonal and plasmid-mediated transmission clusters.** In the study period, there were 58 putative clonal transmission clusters. The median number of acquisition patients per cluster was three (IQR, 2 to 5; maximum, 22). The median duration between detection of the first isolate to the final isolate in the transmission cluster was 97 days (IQR, 12.5 to 246; maximum, 966). Forty-four (75.9%) of the clusters involved patients admitted to more than one hospital. Sixteen (27.6%) of the clusters involved CPE detected in more than one hospital. Of the 58 clusters, 31 (53.5%) were *K. pneumoniae* clusters, 11 (19.0%) were *E. coli* clusters, nine (15.5%) were *E.cloacae* clusters, and seven (12%) were of other species (three *C. freundii*, two *Klebsiella oxytoca*, one *Enterobacter aerogenes*, one *Citrobacter amalonaticus*). With regards to carbapenemase gene classification, 27 clusters (46.6%) were $bla_{NDM-1}$, 22 (37.9%) were $bla_{KPC-2}$, eight clusters (13.8%) were of other carbapenemase genes (two $bla_{OXA-48}$, two $bla_{OXA-232}$, and one cluster each with $bla_{OXA-181}$, $bla_{NDM-7}$, $bla_{IMP-1}$ and $bla_{IMI-1}$) and one cluster (1.7%) was with co-carriage of $bla_{KPC-2}$/$bla_{NDM-1}$.

Sixteen putative plasmid-mediated transmission clusters were detected during the study period. The median number of acquisition patients per plasmid-mediated transmission cluster was five (IQR, 3–9; maximum, 182). The median duration between detection of the first isolate to the final isolate in the plasmid-mediated transmission cluster was 667 days (IQR, 373–908; maximum, 1253). Fourteen (87.5%) of the plasmid-mediated transmission clusters involved patients admitted to more than one hospital. Fourteen (87.5%) of the clusters involved CPE detected in more than one hospital. Fourteen (87.5%) of the plasmid-mediated transmission clusters involved more than one bacterial species. Of the plasmid-mediated transmission clusters, 15 (93.8%) involved *K. pneumoniae*, 12 (75.0%) involved *E. coli*, 10 (62.5%) involved *E. cloacae*, six (37.5%) involved *C. freundii*, three (18.8%) involved *E. aerogenes*, three (18.8%) involved *K. oxytoca*, three (18.8%) involved *Citrobacter koseri*, two (12.5%) involved *C. amalonaticus*, one (6.3%) involved *Citrobacter farmeri*, one (6.3%) involved *Citrobacter rodentium* and one (6.3%) involved *Morganella morganii*. As for carbapenemase gene classification, four clusters (25.0%) were $bla_{NDM-1}$, and there was one cluster each for $bla_{IMP-4}$, $bla_{KPC-2}$, $bla_{NDM-5}$, $bla_{NDM-7}$, $bla_{OXA-181}$ and $bla_{OXA-232}$. Six clusters (37.5%) involved patients co-carrying carbapenemase genes (two clusters of $bla_{NDM-1}$/$bla_{OXA-48}$ and one cluster each for $bla_{NDM-1}$/$bla_{OXA-181}$, $bla_{NDM-1}$/$bla_{OXA-232}$, $bla_{NDM-5}$/$bla_{OXA-181}$ and $bla_{KPC-2}$/$bla_{NDM-1}$/$bla_{OXA-48}$).

**Time trends of putative clonal and plasmid-mediated transmissions.** The total patient-days for participating sites was 8,415,683. Overall incidence of putative clonal transmission increased at a rate of 0.021 (95%CI, 0.015–0.027) acquisition patients per 10,000 patient-days per month, from September 2010 to March 2014 (95% bootstrap confidence interval [BCI], December 2013 to June 2014). Thereafter, until April 2015, incidence of putative clonal transmission decreased at a rate of –0.026 (95%CI, –0.049 – –0.007) acquisition patients per 10,000 patient-days per month. Overall putative plasmid-mediated transmission, on the other hand,

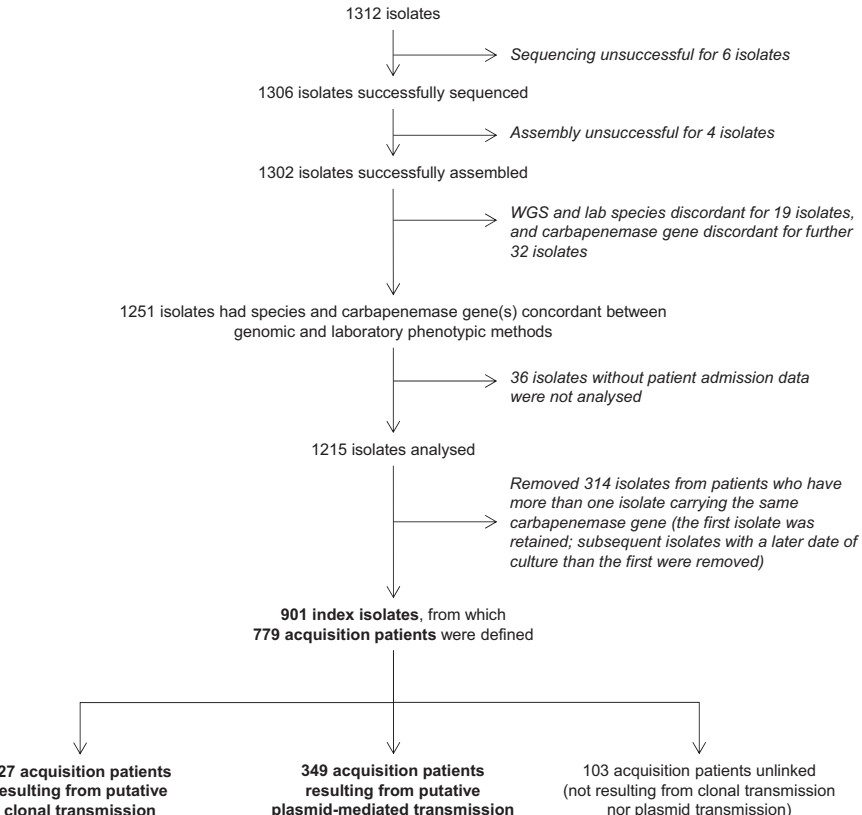

**Fig. 1 Disposition of isolates analysed in the study.** From September 2010 to April 2015, 1312 carbapenemase-producing *Enterobacterales* isolates were obtained from the National Public Health Laboratory, of which 1215 isolates had species and carbapenemase gene(s) concordant between genomic and laboratory phenotypic methods as well as metadata available. Isolates from patients who had more than one isolate carrying the same carbapenemase gene were removed ($N = 314$). A total of 901 index isolates formed the dataset for establishing putative clonal or plasmid-mediated transmission in terms of source-acquisition case pairs, from which 779 acquisition patients were defined.

increased from September 2010 at a rate of 0.016 (95%CI, 0.011–0.024) acquisition patients per 10,000 patient-days per month, with no statistically significant decreasing trend identified during the study period (Fig. 2a, Supplementary Tables 5, 6). Similarly, among surveillance cultures only, the time trend of putative clonal transmission peaked in April 2014 (95%BCI, January 2014 to July 2014) after increasing at a rate of 0.017 (95%CI, 0.011–0.024) acquisition patients per 10,000 patient-days per month from September 2010, and declined thereafter at a rate of –0.035 (95%CI, –0.060 – –0.010) acquisition patients per 10,000 patient-days per month, until April 2015. There was no appreciable decrease noted in incidence of putative plasmid-mediated transmission for surveillance cultures; the upward trend continued at a rate of 0.013 (95%CI, 0.008–0.019) acquisition patients per 10,000 patient-days per month until the end of the study period (Fig. 2b, Supplementary Tables 5, 6). Among clinical cultures, putative clonal and plasmid-mediated transmission increased from September 2010 at a rate of 0.003 (95%CI, 0.002–0.006) and 0.003 (95%CI, 0.002–0.005) acquisition patients per 10,000 patient-days per month respectively, and both did not demonstrate any statistically significant decreases (Fig. 2c, Supplementary Tables 5, 6).

In analysing acquisition time trends of putative clonal transmission and plasmid-mediated transmission stratified by hospital or ward contact, only clonal transmission related to direct ward contact demonstrated a statistically significant downturn with an increase prior to December 2013 (95%BCI, July 2013 to March 2014), at a rate of 0.008 (95%CI, 0.004–0.012) acquisition patients per 10,000 patient-days per month, and a decrease in incidence thereafter at a rate of –0.017 (95%CI, –0.029 – –0.005) acquisition patients per 10,000 patient-days per month. In contrast, direct

ward contact-related plasmid-mediated transmission continued to increase (rate, 0.009 acquisition patients per 10,000 patient-days per month; 95%CI, 0.006–0.015) until the end of the study; furthermore, indirect ward contact-related clonal and plasmid-mediated transmission, and both clonal and plasmid-mediated transmission related to direct and indirect hospital contact did not downtrend (Fig. 2d, e, Supplementary Tables 5, 6). In the second half of the study period, the rate of surveillance cultures, collected as part of infection prevention measures, increased steadily from 19.4 acquisition patients per 10,000 patient-days per month (95% BCI, 18.9–19.9) in June 2013 to 153.5 acquisition patients per 10,000 patient-days per month (95%BCI, 150.3–156.7) in April 2015.

**Epidemiologic and microbiologic risk factors of clonal and plasmid-mediated transmissions.** To determine epidemiologic risk factors of clonal and plasmid-mediated transmissions, 1451 putative clonal transmission case pairs were compared with 1451 available control pairs, and 30,059 putative plasmid-mediated transmission case pairs with 30,059 available control pairs. The number of case pairs (clonal and plasmid-mediated transmission) was higher than the number of acquisition patients as it is possible for an acquisition patient to have multiple potential source patients and these would form separate case pairs. Risk factors associated with clonal transmission in the multivariable analysis were direct ward contact (adjusted odds ratio [aOR], 6.22; 95%CI, 3.89–9.95; $P < 0.0001$), indirect ward contact (aOR, 2.90; 95% CI, 1.89–4.45; $P < 0.0001$), direct hospital contact (aOR, 4.66; 95% CI, 3.20–6.79; $P < 0.0001$) and indirect hospital contact (aOR,

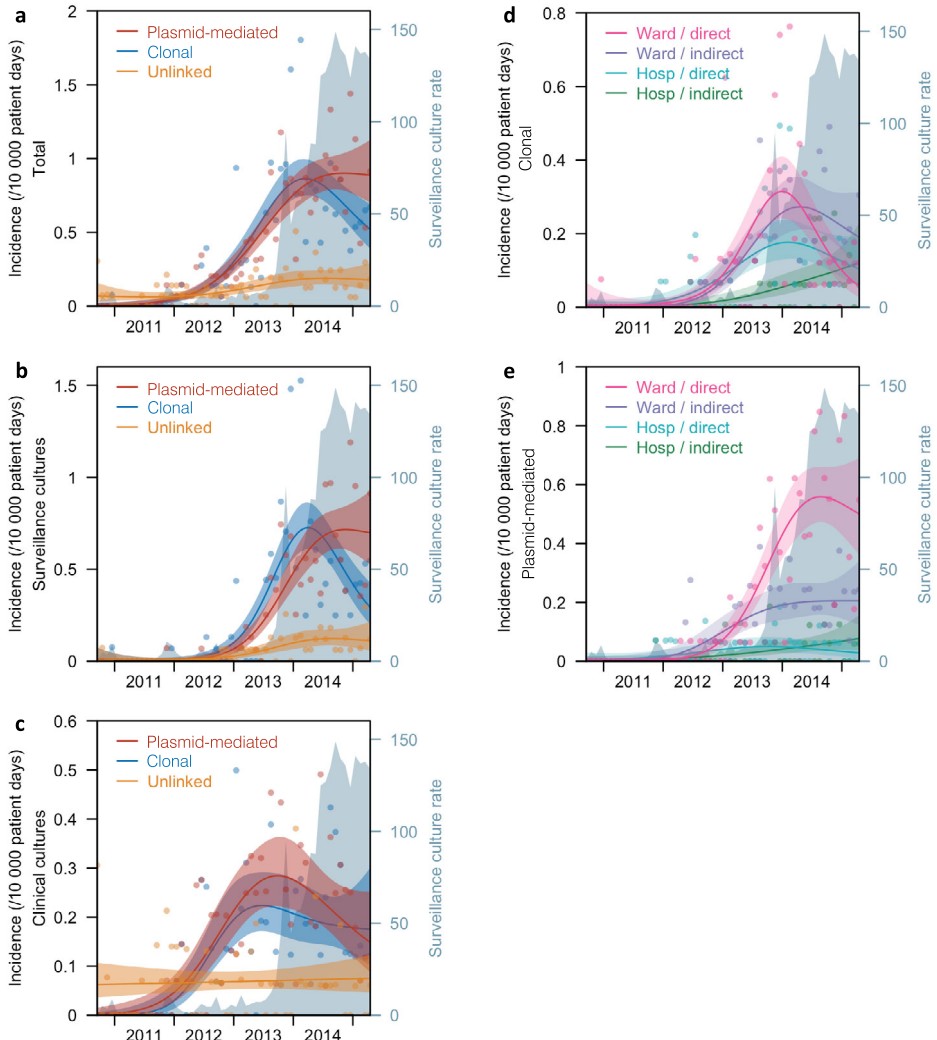

**Fig. 2 Incidence of carbapenemase-producing *Enterobacterales* patients from 2010 to 2015.** The data is presented as (**a**) total incidence of patients resulting from putative clonal or plasmid-mediated transmission, or neither, **b** Incidence of patients among surveillance cultures only, **c** Incidence of patients among clinical cultures only, **d** Incidence of patients resulting from putative clonal transmission stratified by hospital or ward contact, **e** Incidence of patients resulting from putative plasmid-mediated transmission stratified by hospital or ward contact. Incidence is defined as the number of new patient pairs per 10,000 patient-days. Surveillance culture rate is defined as the number of surveillance cultures per 10,000 patient-days. Ward contact was further classified as direct or indirect, and in the absence of any form of ward contact, direct or indirect hospital contact. Time trends are presented as six-period (month) moving averages. Lines represent the incidence point estimate and the shaded areas represent the 95% CI of the estimate.

1.62; 95%CI, 1.07–2.47; *P* = 0.023). Discipline and procedure contact were not associated with clonal transmission. Community contact was not analysed as only one of 1451 case pairs (0.07%) and no control pairs had any form of community contact (Table 1).

Compared with *E. coli*, *K. pneumoniae* (aOR, 3.13; 95%CI, 2.14–4.58; *P* < 0.0001) and *Enterobacter spp.* (aOR, 2.29; 95%CI, 1.46–3.61; *P* = 0.0003) were positively associated with clonal transmission. Additionally, bacteria carrying *bla*$_{NDM}$ (aOR, 1.52; 95%CI, 1.04–2.22; *P* = 0.031) and *bla*$_{OXA-type}$ (aOR, 1.81; 95%CI, 1.13–2.91; *P* = 0.014) genotypes had increased odds of clonal transmission compared with *bla*$_{KPC}$ (Table 1).

Risk factors demonstrating a weak association with plasmid-mediated transmission in the multivariable analysis were direct ward contact (adjusted odds ratio [aOR], 2.14; 95%CI, 0.63–7.27; *P* = 0.22), indirect ward contact (aOR, 1.46; 95%CI, 0.69–3.09; *P* = 0.32), direct hospital contact (aOR, 1.79; 95%CI, 0.98–3.27; *P* = 0.058) and indirect hospital contact (aOR, 1.20; 95%CI, 0.62–2.30; *P* = 0.59). However these contact risk factor

associations did not meet criteria for statistical significance at the α level of 0.05. Discipline and procedure contact were not associated with plasmid-mediated transmission. Community contact was not analysed as only 11 of 30,059 case pairs (0.04%) and two of 30,059 control pairs (0.007%) had any form of community contact (Table 2).

In sensitivity analyses restricted to only source isolates which were sampled at least 7 days before genomically-linked acquisition isolates, inferences regarding epidemiologic risk factors remained unchanged (Supplementary Tables 7, 8).

## Discussion

Our findings provide important insights into the dynamics of CPE introduction and dissemination in healthcare facilities which can inform future policies for CPE control. In Singapore, close to 90% of CPE transmission over 4.7 years of the study period were genomically-linked, with an approximately equal prevalence of putative clonal and plasmid-mediated transmissions. After

**Table 1 Factors associated with putative clonal transmission of CPE.**

| Variable | Clonal-transmission pairs (N = 1451) (%) | Control pairs (N = 1451)(%) | Clonal-transmission pairs[a] (weighted %) | Control pairs[a] (weighted %) | Univariable analysis[b] | | Multivariable analysis[b] | |
|---|---|---|---|---|---|---|---|---|
| | | | | | OR (95%CI) | P-value | aOR (95%CI) | P-value |
| **Hospital or ward contact** | | | | | | | | |
| No hospital nor ward contact | 288 (19.9) | 763 (52.6) | 18.5% | 44.9% | ref | - | ref | - |
| Direct ward contact | 134 (9.2) | 5 (0.3) | 14.5% | 0.5% | 7.55 (5.01-11.4) | <0.0001 | 6.22 (3.89-9.95) | <0.0001 |
| Indirect ward contact | 255 (17.6) | 172 (11.9) | 18.1% | 12.6% | 3.11 (2.08-4.66) | <0.0001 | 2.90 (1.89-4.45) | <0.0001 |
| Direct hospital contact (No ward contact) | 385 (26.5) | 100 (6.9) | 29.0% | 9.1% | 4.99 (3.42-7.29) | <0.0001 | 4.66 (3.20-6.79) | <0.0001 |
| Indirect hospital contact (No ward contact) | 389 (26.8) | 411 (28.3) | 19.9% | 32.9% | 1.65 (1.07-2.53) | 0.022 | 1.62 (1.07-2.47) | 0.023 |
| **Discipline contact** | | | | | | | | |
| No contact | 1115 (76.8) | 1261 (86.9) | 74.9% | 84.9% | ref | - | ref | - |
| Direct contact | 127 (8.8) | 20 (1.4) | 12.4% | 2.0% | 2.72 (1.91-3.87) | <0.0001 | 1.19 (0.80-1.77) | 0.38 |
| Indirect contact | 209 (14.4) | 170 (11.7) | 12.7% | 13.2% | 1.10 (0.76-1.60) | 0.61 | 0.97 (0.64-1.45) | 0.87 |
| **Procedure contact** | | | | | | | | |
| No contact | 1438 (99.1) | 1442 (99.4) | 99.6% | 99.0% | ref | - | ref | - |
| Direct contact | 0 | 0 | 0.0% | 0.0% | - | - | - | - |
| Indirect contact | 13 (0.9) | 9 (0.6) | 0.4% | 1.0% | 0.49 (0.06-3.93) | 0.50 | - | - |
| **Bacterial species** | | | | | | | | |
| Escherichia coli | 283 (19.5) | 593 (40.9) | 18.1% | 39.4% | ref | - | ref | - |
| Klebsiella pneumoniae | 819 (56.4) | 395 (27.2) | 57.5% | 30.4% | 3.05 (2.04-4.55) | <0.0001 | 3.13 (2.14-4.58) | <0.0001 |
| Enterobacter spp | 308 (21.2) | 318 (21.9) | 17.9% | 19.6% | 1.75 (1.14-2.71) | 0.011 | 2.29 (1.46-3.61) | 0.0003 |
| Others | 41 (2.8) | 145 (10.0) | 6.5% | 10.5% | 1.31 (0.65-2.65) | 0.45 | 1.25 (0.59-2.64) | 0.56 |
| **Genotypes** | | | | | | | | |
| $bla_{KPC}$ | 443 (30.5) | 591 (40.7) | 33.3% | 42.0% | ref | - | ref | - |
| $bla_{NDM}$ | 664 (45.8) | 599 (41.3) | 48.0% | 39.7% | 1.39 (1.01-1.90) | 0.042 | 1.52 (1.04-2.22) | 0.031 |
| $bla_{OXA-type}$ | 321 (22.1) | 153 (10.5) | 14.7% | 10.5% | 1.55 (1.01-2.38) | 0.043 | 1.81 (1.13-2.91) | 0.014 |
| Others | 23 (1.6) | 108 (7.4) | 4.0% | 7.8% | 0.69 (0.31-1.56) | 0.38 | 0.77 (0.34-1.78) | 0.55 |
| **Community contact[c]** | | | | | | | | |
| No contact | 1450 (100.0) | 1451 (100.0) | - | - | - | - | - | - |
| Same household | 0 | 0 | - | - | - | - | - | - |
| Same zipcode | 1 (0.0) | 0 | - | - | - | - | - | - |

OR Odds-ratio, aOR Adjusted odds-ratio, Ref Reference, - Not applicable

[a]To correct for potential bias from clustering, the prevalence of epidemiologic risk factors were inversely-weighted by a factor of one over the number of case-control pairs with identical source patient and by reducing the sample size to derive standard errors concomitantly.

[b]Univariable and multivariable analyses were conducted based on the weighted percentages using conditional logistic regression on matched case-control pairs. The Wald chi-square test was performed for all the risk factors with an α level of 0.05 (two-sided). No adjustment was made for multiple comparisons.

[c]Community contact was excluded from univariable and multivariable analyses as the frequency of exposure was too low.

**Table 2 Factors associated with putative plasmid-mediated transmission of CPE.**

| Variable | Plasmid-mediated transmission pairs (N = 30059) (%) | Control pairs (N = 30059) (%) | Plasmid-mediated transmission pairs[a] (weighted %) | Control pairs[a] (weighted %) | Univariable analysis[b] | | Multivariable analysis[b] | |
|---|---|---|---|---|---|---|---|---|
| | | | | | OR (95%CI) | P-value | aOR (95%CI) | P-value |
| **Hospital or ward contact** | | | | | | | | |
| No hospital nor ward contact | 8586 (28.6) | 13690 (45.5) | 39.3% | 52.9% | ref | – | ref | – |
| Direct ward contact | 636 (2.1) | 221 (0.7) | 2.5% | 0.6% | 2.16 (0.81–5.76) | 0.12 | 2.14 (0.63–7.27) | 0.22 |
| Indirect ward contact | 5958 (19.8) | 4070 (13.5) | 16.1% | 11.7% | 1.49 (0.78–2.83) | 0.23 | 1.46 (0.69–3.09) | 0.32 |
| Direct hospital contact (No ward contact) | 4883 (16.2) | 2193 (7.3) | 14.9% | 5.9% | 1.87 (1.06–3.32) | 0.032 | 1.79 (0.98–3.27) | 0.058 |
| Indirect hospital contact (No ward contact) | 9996 (33.3) | 9885 (32.8) | 27.3% | 28.9% | 1.21 (0.66–2.22) | 0.53 | 1.20 (0.62–2.30) | 0.59 |
| **Discipline contact** | | | | | | | | |
| No contact | 24166 (80.4) | 26137 (87.0) | 83.2% | 87.8% | ref | – | ref | – |
| Direct contact | 1343 (4.5) | 556 (1.8) | 3.9% | 1.6% | 1.50 (0.85–2.64) | 0.16 | 1.01 (0.53–1.92) | 0.98 |
| Indirect contact | 4550 (15.1) | 3366 (11.2) | 12.9% | 10.6% | 1.14 (0.65–2.00) | 0.64 | 0.96 (0.50–1.84) | 0.90 |
| **Procedure contact** | | | | | | | | |
| No contact | 29791 (99.1) | 29811 (99.2) | 99.4% | 99.4% | ref | – | ref | – |
| Direct contact | 5 (0.0) | 2 (0.0) | <0.1% | <0.1% | – | – | – | – |
| Indirect contact | 263 (0.9) | 246 (0.8) | 0.6% | 0.6% | 0.98 (0.31–3.12) | 0.97 | – | – |
| **Genotypes** | | | | | | | | |
| $bla_{KPC}$ | 16863 (56.1) | 7556 (25.1) | 40.7% | 30.3% | ref | – | ref | – |
| $bla_{NDM}$ | 12222 (40.7) | 15391 (51.2) | 49.9% | 48.3% | 0.88 (0.55–1.39) | 0.58 | 0.92 (0.58–1.47) | 0.73 |
| $bla_{OXA-type}$ | 308 (1.0) | 4888 (15.3) | 6.6% | 14.3% | 0.49 (0.12–1.91) | 0.30 | 0.52 (0.14–1.89) | 0.32 |
| Others | 666 (2.2) | 2224 (7.4) | 2.9% | 7.1% | 0.44 (0.07–2.67) | 0.38 | 0.46 (0.08–2.70) | 0.39 |
| **Community contact[c]** | | | | | | | | |
| No contact | 30048 (100.0) | 30057 (100.0) | – | – | – | – | – | – |
| Same household | 1 (0.0) | 0 | – | – | – | – | – | – |
| Same zipcode | 10 (0.0) | 2 (0.0) | – | – | – | – | – | – |

OR Odds-ratio, aOR Adjusted odds-ratio, Ref Reference, – Not applicable
[a]To correct for potential bias from clustering, the prevalence of epidemiologic risk factors were inversely-weighted by a factor of one over the number of case-control pairs with identical source patient and by reducing the sample size to derive standard errors concomitantly.
[b]Univariable and multivariable analyses were conducted based on the weighted percentages using conditional logistic regression on matched case-control pairs. The Wald chi-square test was performed for all the risk factors with an α level of 0.05 (two-sided). No adjustment was made for multiple comparisons.
[c]Community contact was excluded from univariable and multivariable analyses as the frequency of exposure was too low.

implementation of various control measures consistent with US CDC and WHO guidelines (Fig. 3 and Supplementary Table 1)[11,12], we observed an overall reduction in putative clonal transmission and continued increase in putative plasmid-mediated transmission. The decline in putative clonal transmission appeared to be mainly due to a reduction in transmission events arising from direct ward contact. The independent association of indirect ward and indirect hospital contact with putative clonal CPE transmission suggest the presence of persistent reservoirs of CPE in the hospital environment. Our analysis found a weak association of contact risk factors with putative plasmid-mediated transmission which did not meet criteria for statistical significance.

Our findings suggest that clonal and plasmid-mediated transmission may be differentially affected by existing infection prevention and control measures. In the first few years following the introduction and spread of CPE in Singapore, there was a decrease in clonal transmissions that were associated with direct ward contact. Early detection and isolation of CPE-carriers, as practiced in Singapore, would be expected to decrease onward clonal transmission[13]. Additionally, the aggressive contact tracing strategy employed by most study sites to identify patients with epidemiological links to the index patients, such as sharing a ward, may have contributed to the decreased onward clonal transmission. Detection and isolation of these secondary cases would be expected to prevent further onward clonal transmission.

Putative clonal transmission resulting in patient acquisition through indirect ward and indirect hospital contact did not decrease during the study period. Transmission via indirect contact suggests the presence of unknown and persistent reservoirs of CPE in the hospital apart from known CPE-positive patients. Undetected CPE-positive inpatients are less likely to be the main drivers of indirect transmission as increased surveillance and isolation of detected CPE carriers over the study period would be anticipated to decrease the number of and transmission via silent carriers. Recent evidence shows that hospital-environment microbiomes offer distinct ecological niches for opportunistic pathogens and antibiotic resistant Gram-negative bacteria. Hospital-environment resistomes were found to include multiple carbapenemases, be dynamic in nature, and persist in the hospital environment for extended periods (more than eight years)[14]. In particular, the hospital water environment, including sinks and shower and toileting facilities, have been implicated as potential reservoirs of CPE in the hospital[15].

The lack of association of procedure (gastrointestinal and urologic endoscopy) contact with genomically-linked CPE transmissions suggests that while these routes of transmissions have been documented in focal outbreaks[16], they were not significant routes of CPE transmissions in the study population. Robust microbiological surveillance programmes that were instituted in the study sites, some as early as 2012, may have played a role in preventing active CPE transmissions via endoscopes. Residential and community contact was not a significant risk factor for genomically-linked CPE transmissions even though a recent study has reported household CPE transmission rate of up to 2%[7]. However, a purpose-designed study would be needed to understand the role of household transmission of CPE in Singapore as, in the current study, the overall number of patients sharing the same household was extremely small.

Putative plasmid-mediated transmission did not appreciably decrease during the study period which suggests limited impact of existing infection prevention measures. The weak association of contact risk factors (direct ward contact, indirect ward contact, direct hospital contact and indirect hospital contact) with putative plasmid-mediated transmission suggests that factors other than patient to patient contact may play a major role in carbapenemase plasmid dissemination. Prior studies have shown that conjugation of carbapenemase-encoding plasmids is associated with plasmid persistence in the hospital over prolonged periods, potentially as different species are suited to different ecologic niches[17]. Plasmid conjugation in varied bacterial species may facilitate transmission of the carbapenemase gene from an index patient to the inanimate environment, especially the aqueous environment, and subsequent transmission to another patient. Plasmid-mediated transmission and the contribution of hidden reservoirs to hospital CPE transmission is a domain which requires further study.

Many studies of CPE outbreak investigations have focused on clonal transmission, relying upon single-nucleotide polymorphism distances indicative of clonal bacterial spread to determine genomically-linked transmission[10,18,19]. As plasmid-mediated transmission accounts for a significant prevalence of CPE transmission at a population-level, plasmid identification and linkage would be necessary for accurate CPE outbreak investigation[20]. In this regard traditional whole-genome sequencing methods and analysis which worked well for drug-resistant bacteria spread mainly by clonal transmission, for example MRSA[21], would need to be augmented by robust plasmid linkage methods for CPE transmission.

Our findings in the context of a national network using whole-genome sequencing and detailed hospitalization data enabled us to determine population-level contact associations with the two main genomic modes of CPE transmission and did not depend on limited epidemiologic information, a feature of most single-centre studies[10,22,23]. The inclusion of all CPE isolates, regardless of species, facilitated a more comprehensive picture of plasmid-mediated transmission compared with prior single-species studies[24].

Our study had several limitations. Notably, the rate of surveillance cultures increased substantially over time, and this detection bias would result in overestimation of numbers in the second half of study period. Nevertheless, this would imply that the reduction in putative clonal transmission is likely to be more dramatic, and the dynamics of putative clonal and plasmid-mediated transmission remains evidently different. Private hospitals which provided 20% of inpatient care were excluded from our study. Our findings reflect transmission in a context where the majority of patients are cared for in four- to six-bedder cubicles and this may impact the generalizability to single room-only facilities. As the period of colonization prior to CPE detection is unknown, we were unable to account for transmission during the period preceding CPE detection in the source patient. Related to this, it is possible that acquisition patients could in reality have transmitted to source patients during the period prior to CPE detection in the acquisition patient. However, the sensitivity analysis restricting to source isolates collected at least 7 days prior to genomically-linked acquisition isolates did not alter the epidemiologic risk factors identified. Healthcare worker sampling was not performed; however, there was no statistically significant association between clinical discipline and putative clonal nor plasmid-mediated transmission. We also did not perform environmental sampling and hence were unable to directly confirm the role of the inanimate environment in transmission. Nevertheless, our findings corroborate with other published data from Singapore supporting the notion that inanimate hospital environments could be a source of CPE transmission[14]. Our current study involved only *Enterobacterales* isolates and hence we are unable to directly detect transmission of carbapenemase genes via other gram-negatives, for example, non-fermenters. However our data suggests that some transmission via non-fermenters would be identified as plasmid-mediated transmission. While we did use de novo long-read sequence data paired with short-read data to close reference plasmids locally, plasmid assignments were based on

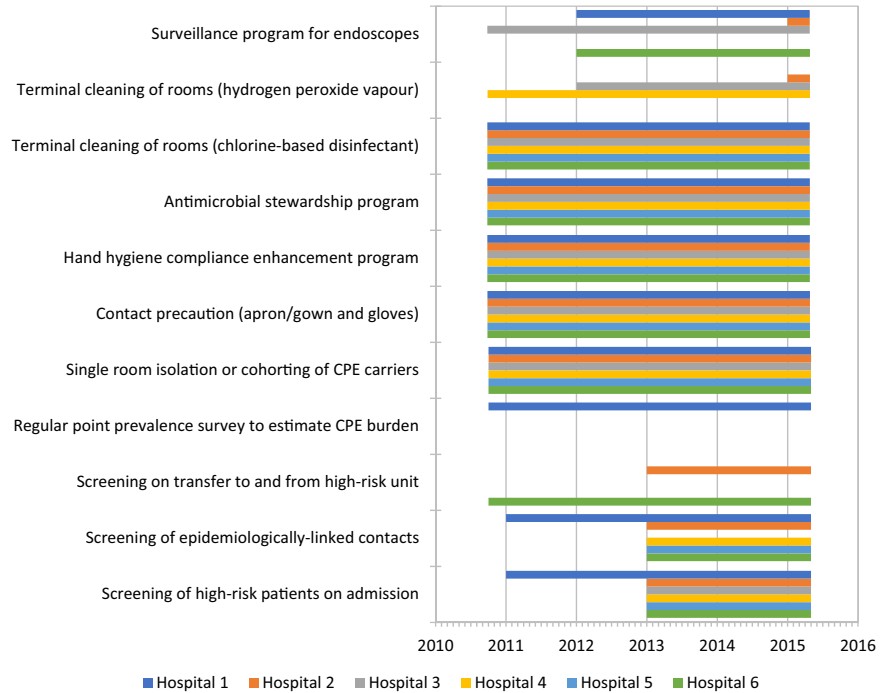

**Fig. 3 Staggered adoption of infection-prevention measures at participating study sites over study period (September 2010 to April 2015).** The coloured-bars represent the time period of adoption of the various measures with calendar years denoted on the x-axis. Specific measures are listed on the y-axis. The bars are colour-coded to differentiate the various participating hospitals.

reference plasmid calling which could result in plasmid mis-identification. Such misclassification, if non-differential, would be expected to underestimate any positive associations with putative plasmid-mediated transmission detected.

In conclusion, while overall CPE incidence did not appreciably decrease, infection prevention measures were associated with statistically significant decreases in direct ward contact putative clonal transmission. Indirect ward and hospital contact transmission, both associated with putative clonal and plasmid-mediated transmissions suggests persistent reservoirs of CPE in the hospital microbiome, possibly the inanimate environment. Plasmid CPE transmission and persistent carbapenemase gene reservoirs in the hospital will need to be considered in future research, surveillance and prevention interventions before effective control of nosocomial CPE transmission prevention can be achieved.

## Methods

**Study design, setting and population.** From September 6, 2010 to April 28, 2015, all six multi-disciplinary public hospitals, providing approximately 80% of inpatient medical care[25] in Singapore (estimated population size, 5.5 million in 2015), participated in this retrospective cohort study. A specialty women's and children's hospital was excluded from the study as the patient population was not representative of the general hospital population in Singapore. Study sites comprised acute, sub-acute, and long-term acute care inpatient facilities with patients occupying single rooms or bays of four to eight beds. The combined inpatient capacity was approximately 9000 beds[26] with capacity per site ranging from 300 to 1600 beds. Two hospitals were academic medical centers with solid organ and stem cell transplant units and four were teaching hospitals with academic affiliations. Throughout the study period, the infection prevention measures at study sites were consistent with US CDC and WHO guidelines[11,12]. Specific infection prevention measures included surveillance cultures for asymptomatic carriers identified via screening of high-risk patients and epidemiologically-linked contacts; regular point prevalence surveys to estimate CPE burden; geographical separation and contact precautions for CPE carriers; enhancement of hand hygiene compliance; anti-microbial stewardship programs; terminal cleaning of rooms occupied by CPE carriers and surveillance programs for endoscopes (Fig. 3 and Supplementary Table 1).

During the study period, all microbiology laboratories in Singapore had submitted carbapenem-resistant *Enterobacterales* (CRE) isolates to the National

Public Health Laboratory (NPHL). NPHL provided all CRE isolates from inpatient clinical and/or surveillance cultures from the study sites, as well as associated metadata (species of the *Enterobacterales*, the types of specimens, and their genotypes) for the study. We extracted the following patient-level factors from the electronic medical records: age, gender, clinical discipline, endoscopic procedures, home address, and inter- and intra-facility movement data. For each patient, we collected all hospital admissions, all ward and bed movements, dates of admissions and dates of discharge during the study period across participating sites.

**Microbiological methods, genomic sequencing and analysis.** CRE isolates suspected of carrying carbapenemase genes were submitted to NPHL by participating microbiology laboratories for further phenotypic characterization and polymerase chain reaction-based assays. CPE isolates were obtained from NPHL for whole genome sequencing. DNA was extracted and sequenced with the use of Illumina technology[2].

Bacterial core genome analysis was based on a previously published pipeline (Supplementary Fig. 2)[24]. Putative clonal transmission was identified if two isolates had the same sequence type-cluster (Supplementary Figs. 1 and 2, and Supplementary Table 2), same carbapenemase gene allele and a pairwise single nucleotide polymorphism (SNP) value below the BEAST-derived mutation rate threshold[27] (Supplementary Table 3).

Plasmid identification was performed for all isolates using PlasmidSeeker (version 0.1)[28] against the carbapenemase gene allele-specific reference databases (Supplementary Table 4). Subsequently, putative plasmid-mediated transmission was established between two isolates if they shared at least one plasmid carrying a carbapenemase gene.

Details regarding genomic sequencing and analysis are reported in the Methods section of the Supplementary Information.

**Clinical epidemiology.** An index isolate was the first-detected isolate carrying a carbapenemase gene in a patient during the study period. An acquisition patient, defined as the patient from which an index isolate was obtained, may have multiple index isolates if more than one CPE isolate had the same date of culture. Transmission was considered to have occurred if a source isolate from a different patient with an earlier or same date of culture (source patient) could be genomically-linked (either by putative clonal transmission or putative plasmid-mediated transmission) to the index isolate of the acquisition patient. Acquisition patients were classified as arising from clonal transmission (as long as putative clonal transmission criteria was fulfilled), plasmid-mediated transmission (if only putative plasmid-mediated transmission criteria was fulfilled), or unlinked (neither criteria fulfilled). The risk period for transmission was defined as the interval between the sampling dates for the source and acquisition isolates.

A putative clonal transmission cluster was defined as acquisition patients which met putative clonal transmission criteria with at least one other acquisition patient in the cluster. Similarly, a putative plasmid-mediated transmission cluster was defined as acquisition patients which did not meet putative clonal transmission criteria but did meet putative plasmid-mediated transmission criteria with at least one other acquisition patient in the cluster.

To determine epidemiologic risk factors for CPE transmission, source and acquisition patients who fulfilled criteria for putative clonal transmission or plasmid-mediated transmission were classified as source-acquisition patient pairs. Epidemiologic relationships between each patient pair were analysed in terms of community, hospital, discipline, and procedure contact. Community contact was defined as (i) household level, if the patient pair belonged to the same household; (ii) zipcode level, if the patient pair were not from the same household but the residential addresses of the source and acquisition patients had the same zipcode. Hospital contact was defined as: (i) direct ward contact, if the patient pair was admitted to the same ward, overlapping for at least one calendar day within the risk period; (ii) indirect ward contact, if the patient pair was admitted to the same ward, but the period the source patient resided in the ward preceded that of the acquisition patient without overlap within the risk period; (iii) direct hospital contact, if the patient pair was admitted to the same hospital, although not the same ward, and the inpatient period of the source patient and the acquisition patient overlapped for at least one day within the risk period; (iv) indirect hospital contact, if the patient pair was admitted to the same hospital, although not the same ward, and the inpatient period of the source patient preceded that of the acquisition patient without overlap within the risk period; and (v) no hospital contact. Discipline contact was defined as: (i) direct, if the source patient and the acquisition patient were admitted under the same clinical discipline in the same hospital and overlapped for at least one day within the risk period; (ii) indirect, if the source patient and acquisition patient were admitted under the same clinical discipline in the same hospital but the period of admission of the source patient preceded that of the acquisition patient without overlap within the risk period; and (iii) no discipline contact. Procedure contact was defined as: (i) direct, if the same procedure was performed on a patient pair in the same hospital on the same date within the risk period; (ii) indirect, if the same procedure was performed on a patient pair in the same hospital within the risk period, but the procedure date for the source patient preceded that of the acquisition patient; and (iii) no procedure contact.

For the analysis of associations to determine epidemiologic contact risk factors for clonal and plasmid-mediated transmissions, control acquisition patients were selected to form control pairs for each source-acquisition patient pair (case pair). For a given putative clonal transmission case pair, control acquisition patients were randomly selected from all acquisition patients who met the following criteria: (i) the control acquisition patient was not the same as the acquisition patient or source patient of the case pair; (ii) the date of culture of the isolate from the control acquisition patient was same as or later than the date of culture of the isolate from the acquisition patient in the case pair; and (iii) the isolates of the control acquisition patient did not meet putative clonal transmission criteria with the source patient or the acquisition patient in the case pair. For a given putative plasmid-mediated transmission case pair, controls were randomly selected from all acquisition patients who meet the following criteria: (i) the control acquisition patient was not the same as the acquisition patient or source patient of the case pair; (ii) the date of culture of the isolate from the control acquisition patient was same as or later than that from the acquisition patient in the case pair; (iii) the isolates of the control acquisition patient did not meet putative clonal or plasmid-mediated transmission criteria with the source patient or acquisition patient in the case pair.

**Statistical analysis**. Incidence over time was determined based on the number of acquisition patients and estimated using thin-plate regression splines using Wood's method[29] in R[30]. The rate of change was defined as the number of acquisition patients per 10,000 patient-days per month. Confidence intervals for peak incidence were derived by nesting the spline models within a bootstrap, taking months as the sampling unit: upper bounds coinciding with the last month of data collection (April 2015) indicated lack of statistically significant evidence of a turning point in the study period. Bootstrapping was performed using sampling with replacement with 1000 replicates. Rates of increase or decrease were calculated over the duration of the study from the periods before and after the estimated peak incidence from the splines and similarly bootstrapped to obtain growth and decline rates, respectively.

Conditional logistic regression on all matched case-control pairs as defined above was implemented to assess the association between epidemiologic contact risk factors and putative clonal and plasmid-mediated transmissions. To correct for potential bias due to clustering of epidemiologic contact risk factors when an identical source patient generated multiple case-control pairs, the prevalence of associated epidemiologic risk factors in these situations were inversely weighted by a factor of one over the number of case-control pairs generated by the identical source patient and by reducing the sample size to derive standard errors concomitantly. We selected variables representative of different potential modes of CPE transmission for multivariable regression and included variables with exposure prevalence more than 10%, greater effect size on univariable analysis and which were statistically significant ($p < 0.05$). The Wald chi-square test was performed for all the risk factors with an α level of 0.05 (two-sided). P-values were interpreted together with 95% confidence interval (CI) for the odds ratios and adjusted odds ratios derived from the conditional logistic regression model. R[30] was used for statistical analysis unless otherwise specified.

**Sensitivity analysis**. As acquisition patients in our analysis could possibly have transmitted to patients classified as source patients, a sensitivity analysis was performed to reduce this possibility and examine the impact on the epidemiologic risk factors identified by restricting source isolates to those sampled at least 7 days before genomically-linked acquisition isolates.

**Ethics and reporting**. The study was reviewed and approved by the ethics institutional review boards of National Health Group Singapore (DSRB reference: 2014/00617) which did not require that patients provide written informed consent. This retrospective cohort study adhered to the Strengthening the Reporting of Observational Studies in Epidemiology (STROBE) statement guidelines for reporting observational studies[31]. The corresponding author attests that the authors had access to all the study data, take responsibility for the accuracy of the analysis, and had authority over manuscript preparation and the decision to submit the manuscript for publication. The corresponding author had final responsibility for the decision to submit for publication.

**Reporting summary**. Further information on research design is available in the Nature Research Reporting Summary linked to this article.

## Data availability

All raw sequence data have been uploaded to the NCBI Sequence Read Archive Database (Bioproject Accession Numbers: PRJNA757551 and PRJNA765801 for the Illumina short-read sequencing data and PRJNA801415 for the Oxford Nanopore Technologies long-read sequencing data that contributed to the plasmid genome sequence reference database described in the Supplementary Information). The Beta-Lactamase Database was downloaded from NCBI (https://www.ncbi.nlm.nih.gov/pathogens/beta-lactamase-data-resources/; Date of download: May 2017). Species-specific MLST allelic profiles were downloaded from PubMLST (https://pubmlst.org/data/; Date of download: 10 January 2018 for K. pneumoniae, 13 March 2018 for E. coli and E. cloacae, 17 August 2018 for C. freundii and K. oxytoca). The NCBI RefSeq Genome Database was obtained from NCBI (https://www.ncbi.nlm.nih.gov/refseq/; Date of download: July 2018). A detailed listing of the isolates analysed (e.g. species, ST, carbapenemase gene, sampling site) is available in Supplementary Data 1. The epidemiological/ward movement data are protected and are not available due to data privacy laws. Data can be requested by emailing the corresponding authors with an expected maximum timeframe of reply of 3 months. Data use agreements may be required depending on the specific nature of request. All other data that support the findings of this study and a detailed description of the methods used are available in the manuscript or in the Supplementary Information.

## Code availability

A detailed description of the tools used is available in the manuscript or in the Supplementary Information. The custom code used in the analysis is made available on Github at https://github.com/nataschamay/cp_transmission_2021 as well as on Zenodo at https://doi.org/10.5281/zenodo.6363989, with related details provided in Supplementary Information[32]. Statistical programming code is provided in Supplementary Software 1.

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

## Acknowledgements

We thank the Singapore Infectious Diseases Initiative, Infection Prevention and Control units of contributing hospitals, Singapore Clinical Research Network (SCRN), and the Singapore Clinical Research Institute (SCRI). This research was supported by the Singapore Ministry of Health's National Medical Research Council (NMRC) under its two NMRC Collaborative Grants: Collaborative Solutions Targeting Antimicrobial Resistance Threats in Health Systems (CoSTAR-HS) (CGAug16C005 and CG21APR2005; O.T.N.), NMRC Clinician Scientist Award (MOH-000276; O.T.N.), NMRC Clinician Scientist Individual Research Grant (MOH-CIRG18nov-0006; K.M.) and NMRC COVID-19 Research Funds (MOH-000469 and MOH-000717; O.T.N.). Additional support was provided by the German Federal Ministry of Health (BMG) COVID-19 Research and development funding to WHO (Award number 70826; K.M.) and the National Centre for Infectious Diseases (NCID) Catalyst Grant (FY202013VKHQ; V.K.). D.W.E. is a Robertson Foundation Fellow. The funding agencies had no role in the design of the study; the collection, analysis, and interpretation of the data; and the decision to approve publication of the finished manuscript. Any opinions, findings and conclusions or recommendations expressed in this material are those of the author(s) and do not reflect the views of the Singapore Ministry of Health, NMRC and/or NCID.

## Author contributions

K.M. and O.T.N. conceived of and led the study. I.V., B.P.Z.C., R.K.C.F., S.K.P., S.T.O., N.S., K.C.T., T.H.K., P.P.D., T.Y.T., D.C., R.N.D., N.W.S.T., A.K., Y.C., Y.-Y.T., M.A., R.T.P.L., and J.T. were involved in epidemiological and microbiological data collection from the participating institutions. K.M., O.T.N., V.K., N.M.T., S.R.S.P., W.X., W.X.K., and A.R.C. were involved in data analysis, data interpretation, and writing of the manuscript in consultation with S.H., E.P., P.A.T., L.Y.H., D.H., N.S., D.W.E., D.C., and A.C.

## Competing interests

D.W.E. declares personal fees from Gilead outside of the submitted work. The remaining authors declare no competing interests.

## Additional information

[1]National Centre for Infectious Diseases, Singapore, Singapore. [2]Tan Tock Seng Hospital, Singapore, Singapore. [3]Yong Loo Lin School of Medicine, National University of Singapore and National University Health System, Singapore, Singapore. [4]Singapore General Hospital, Singapore, Singapore. [5]Infection Control Program, WHO Collaborating Center, Geneva University Hospitals and Faculty of Medicine, Geneva, Switzerland. [6]Departments of Internal Medicine and Epidemiology, University of Iowa Carver College of Medicine, Iowa City, USA. [7]Center for Comprehensive Access & Delivery Research and Evaluation, Iowa City Veterans Affairs Medical Center, Iowa City, USA. [8]DUKE-NUS Medical School, National University of Singapore, Singapore, Singapore. [9]Changi General Hospital, Singapore, Singapore. [10]Ng Teng Fong General Hospital, Singapore, Singapore. [11]Khoo Teck Puat Hospital, Singapore, Singapore. [12]National University Hospital and National University Health System, Singapore, Singapore. [13]KK Women's and Children's Hospital, Singapore, Singapore. [14]Saw Swee Hock School of Public Health, National University of Singapore and National University Health System, Singapore, Singapore. [15]National University of Singapore, Singapore, Singapore. [16]Academic Clinical Programme (Medicine), SingHealth Duke-NUS, Singapore, Singapore. [17]Program in Emerging Infectious Diseases, Duke-NUS Medical School, Singapore, Singapore. [18]Clinical Centre, National Institutes of Health, Bethesda, USA. [19]Nuffield Department of Medicine, University of Oxford, Oxford, UK. [20]Nuffield Department of Population Health, University of Oxford, Oxford, UK. [21]Lee Kong Chian School of Medicine, Nanyang Technological University, Singapore, Singapore. *A list of authors and their affiliations appears at the end of the paper.
✉email: kalisvar_marimuthu@ttsh.com.sg; oon_tek_ng@ncid.sg

## Carbapenemase-Producing Enterobacteriaceae in Singapore (CaPES) Study Group

Kalisvar Marimuthu[1,2,3], Indumathi Venkatachalam[4], Benjamin Pei Zhi Cherng[3,4,8], Raymond Kok Choon Fong[9], Surinder Kaur Pada[10], Say Tat Ooi[11], Nares Smitasin[3,12], Koh Cheng Thoon[13], Li Yang Hsu[3,14,15], Tse Hsien Koh[4,8], Partha Pratim De[2], Thean Yen Tan[9], Douglas Chan[10], Rama Narayana Deepak[11], Nancy Wen Sim Tee[12], Michelle Ang[1], Raymond Tzer Pin Lin[1,12], Jeanette Teo[12] & Oon Tek Ng[1,2,21]

