## [Peer Review File · Nature Communications]

Whole genome sequencing reveals hidden transmission of carbapenemase-producing *Enterobacterales*Editorial Note: This manuscript has been previously reviewed at another journal that is not operating a transparent peer review scheme. This document only contains reviewer comments and rebuttal letters for versions considered at *Nature Communications*. Mentions of the other journal have been redacted.

REVIEWER COMMENTS

Reviewer #4
Please see attached pdf

Reviewer #5 (Remarks to the Author):

Dear Editor,

thank you for your invitation to review this manuscript. I apologize for my late reply. I have carefully looked into the materials and methods and in particular the genomic sequencing. I have no concerns regarding the methods used. The applied methods are very good, threshold settings for selection of carba alleles are appropriate. Combination of short-read and long-read sequencing for plasmid assemblies are good. core genome analysis is very good. Number of plasmids included in the study is very high. Overall, very thorough analyses. Paper is very well written, reads well.

Review for the paper by Marimuthu et al. for
[REDACTED]

June 1, 2021

1 Focus of the review

The review focuses on the following details:

- Epidemiological study design
- Particularly: the selection of controls
- Statistical methods

2 General comments

1312 CPE isolates, in 817 patients were the basis of the analysis, the final number of patients went down to 779 acquisition subjects for reasons of missing values in covariates. Across these 779 patients, evaluations about clonal transmission, plasmid mediated transmission, or unlinked transmission were done. I cannot comment on the complexity of this task, but if there was uncertainty about the three categories, this could affect the results. I consider these three subgroups of transmission categories as correctly classified. Reasons for collecting the cultures were for surveillance or for clinical reasons. This differentiation could be looked at in more detail, as a sensitivity analysis. The median duration between detection of the first isolate to the final isolate was 97 days, which seems long.

For me, details of the data pre-processing, selection of patients into the analysis, as well as statistical analysis are not clear and I would ask the authors to clarify.

2.1 Data pre-processing

- Aim of this step was to combine each so-called source isolate (patient) with one or more acquisition isolates (patients). These form a so-called case pair. It is possible for an acquisition patient to have multiple source

patients / isolates. Could it be that patients which you consider acquisition subjects have passed the transmission on, rather than the source subjects, they are linked to?

- To find so-called control pairs, the same source patient was combined with a non-acquisition patient. Certain criteria for these pairings are described.
- The process of finding suitable acquisition patients for source patients is a methodologically challenging and potentially algorithmic task, especially if this was done for different situations in which the source and the acquisition patients' ways could have crossed during their hospital stay or even out of hospital. I would like to see statistical code to achieve the task. The completion of the task is even more challenging for the control pairings, so coding would be informative here, as well.

2.2 Data analysis

- The authors describe the methodology for data analysis, as being conditional logistic regression models to account for the paired case-control design. The authors report that 58 clonal transmission clusters and 16 plasmid-mediated transmission clusters were identified. In the subsequent analyses, the underlying number of patients seems very large ($n=1451$) but the actual number of observations is lower. Do source patients enter twice in the analysis? How does the model account for the correlation structure? There is an additional weighting, but given the information I have, I cannot judge if this captures the nested correlation structure. Double use of source patients is problematic. Please clarify.
- General approach for data analysis: the control pairings are selected under certain conditions, but these may lead to a circular reasoning given the transmission locations, potentially. Could the authors please comment on that?
- Please provide statistical programming code and if possible also the data for review.

Reply: We thank the reviewer for the comments and attach our replies below. Line numbers provided are referring to the version with Track Changes activated.

1 Focus of the review

The review focuses on the following details:

- Epidemiological study design
- Particularly: the selection of controls
- Statistical methods

2 General comments

1312 CPE isolates, in 817 patients were the basis of the analysis, the final number of patients went down to 779 acquisition subjects for reasons of missing values in covariates. Across these 779 patients, evaluations about clonal transmission, plasmid-mediated transmission, or unlinked transmission were done. I cannot comment on the complexity of this task, but if there was uncertainty about the three categories, this could affect the results. I consider these three subgroups of transmission categories as correctly classified.

Reasons for collecting the cultures were for surveillance or for clinical reasons. This differentiation could be looked at in more detail, as a sensitivity analysis.

Reply:

We have performed sensitivity analyses to determine the estimates of clonal transmission and plasmid-mediated transmission limiting to surveillance cultures only and clinical cultures only. The findings have been added in the text (lines 81-86):

“Considering only surveillance cultures for infection control purposes, there were 525 acquisition subjects, of which 186 (35.4%) met criteria for clonal transmission, 231 (44.0%) met plasmid-mediated transmission criteria and 108 (20.6%) were unlinked. Considering only clinical cultures, there were 348 acquisition subjects, of which 93 (26.7%) met criteria for clonal transmission, 135 (38.8%) met plasmid-mediated transmission criteria and 120 (34.5%) were unlinked.”

In the current manuscript we have performed time trend analysis based on limiting acquisition isolates to surveillance isolates and clinical isolates. The results are shown in Figure 2B and 2C, Supplementary Table 6 as well as described in the text (lines 145 to 158):

“Similarly, among surveillance cultures only, the time trend of clonal transmission peaked in April 2014 (95%BCI, January 2014 to July 2014) after increasing at a rate of 0.017 (95%CI, 0.011—0.024) acquisition subjects per 10,000 patient-days per month from September 2010, and declined thereafter at a rate of -0.035 (95%CI, -0.060 – -0.010) acquisition subjects per 10,000 patient-days per month, until April 2015. There was no appreciable decrease noted in plasmid-mediated transmission incidence for surveillance cultures; the upward trend continued at a rate of 0.013 (95%CI, 0.008—0.019) acquisition subjects per 10,000 patient-days per month until the end of the study period (Figure 2B, Supplementary Tables 5 and 6). Among clinical cultures, clonal and plasmid-mediated transmission increased from September 2010 at a rate of 0.003 (95%CI, 0.002—0.006) and 0.003 (95%CI, 0.002—0.005) acquisition subjects per 10,000 patient-days per month respectively, and both did not

demonstrate any statistically significant decreases (Figure 2C, Supplementary Tables 5 and 6).”

The median duration between detection of the first isolate to the final isolate was 97 days, which seems long.

Reply:

The median duration of 97 days in a clonal transmission cluster is consistent with prior published data about the length of such difficult-to-control outbreaks.

A single-centred KPC-positive *Klebsiella pneumoniae* clonal outbreak at the Clinical Centre at the US National Institutes of Health lasted at least from June 2011 to December 2011 (approximately 6 to 7 months) highlighting the potential for carbapenemase-positive *Enterobacterales* to spread clonally for months (Ref: Snitkin ES, et al; Tracking a hospital outbreak of carbapenem-resistant *Klebsiella pneumoniae* with whole-genome sequencing. *Sci Transl Med.* 2012).

Similarly, whole-genome sequencing revealed ongoing transmission of KPC-3 positive *E. cloacae* in a burns unit for at least 2 years (Ref: Kanamori H, et al. A Prolonged Outbreak of KPC-3-Producing *Enterobacter cloacae* and *Klebsiella pneumoniae* Driven by Multiple Mechanisms of Resistance Transmission at a Large Academic Burn Center. *Antimicrob Agents Chemother.* 2017)

For me, details of the data pre-processing, selection of patients into the analysis, as well as statistical analysis are not clear and I would ask the authors to clarify.

2.1 Data pre-processing

- Aim of this step was to combine each so-called source isolate (patient) with one or more acquisition isolates (patients). These form a so-called case pair. It is possible for an acquisition patient to have multiple source patients / isolates. Could it be that patients which you consider acquisition subjects have passed the transmission on, rather than the source subjects, they are linked to?

Reply:

Thanks a lot for this pertinent comment. Indeed, it is possible for an acquisition patient to have multiple potential source patients and these would form separate case pairs.

We acknowledge that patients considered acquisition subjects could have passed the transmission on rather than the source patients they were linked to. The study design minimised this by considering the observed timelines and allowing source isolates to only have earlier or same dates of sampling as the genomically-linked acquisition isolate.

Misclassification of acquisition and source subjects resulting from acquisition subjects passing on the infection would affect the epidemiologic risk factor analysis. As such we conducted a sensitivity analysis increasing the time interval to only allow source isolates which were sampled 7 days or more before the genomically-linked acquisition isolate to further decrease the possibility that acquisition patients passed on the infection to source patients. The associations remained unchanged in this sensitivity analysis.

The edited sections related to this are:

In the Methods section (lines 465 to 469):

“Sensitivity analysis

As acquisition subjects in our analysis could possibly have transmitted to subjects classified as source subjects, a sensitivity analysis was performed to reduce this possibility and examine the impact on the epidemiologic risk factors identified by restricting source isolates to those sampled at least 7 days before genomically-linked acquisition isolates.”

In the Results section (lines 208 to 210):

“In sensitivity analyses restricted to only source isolates which were sampled at least 7 days before genomically-linked acquisition isolates, inferences regarding epidemiologic risk factors remained unchanged (Supplementary Tables 7 and 8).”

- To find so-called control pairs, the same source patient was combined with a non-acquisition patient. Certain criteria for these pairings are described.
- The process of finding suitable acquisition patients for source patients is a methodologically challenging and potentially algorithmic task, especially if this was done for different situations in which the source and the acquisition patients' ways could have crossed during their hospital stay or even out of hospital. I would like to see statistical code to achieve the task. The completion of the task is even more challenging for the control pairings, so coding would be informative here, as well.

Reply:

We have detailed the steps for selecting source patients for each acquisition subject to form clonal transmission case pairs and also included the relevant code as part of our submission (Supplementary information on custom code). Similarly, we have also detailed the steps for selecting control pairs for each clonal transmission case pair and also included the relevant code as part of our submission (Supplementary information on custom code).

2.2 Data analysis

- The authors describe the methodology for data analysis, as being conditional logistic regression models to account for the paired case-control design. The authors report that 58 clonal transmission clusters and 16 plasmid-mediated transmission clusters were identified. In the subsequent analyses, the underlying number of patients seems very large (n= 1451) but the actual number of observations is lower. Do source patients enter twice in the analysis? How does the model account for the correlation structure? There is an additional weighting, but given the information I have, I cannot judge if this captures the nested correlation structure. Double use of source patients is problematic. Please clarify.

Reply:

Thank you for this comment—after careful reflection we have modified the analyses to ensure the sample size is adequately characterised in the revised analysis, so we are very grateful for the request for clarification. Source patients could enter twice in the conditional logistic regression to determine association between epidemiologic contact risk factors and clonal and plasmid-mediated transmissions. In the analysis in the original submission, this was partly addressed by down-weighting case-control pairs if there are multiple potential case-control pairs. The problem with that approach, which the reviewer's comment alerted us to, is that in the calculations the sample size should also be reduced concomitantly. This was not accounted for in the original submission. We have fixed this in the revision. So doing has affected the results, in particular on the plasmid-mediated transmissions.

As stated in our methods (lines 452 to 457), "*To correct for potential bias due to clustering of epidemiologic contact risk factors when an identical source subject generated multiple case-control pairs, the prevalence of associated epidemiologic risk factors in these situations were inversely weighted by a factor of one over the number of case-control pairs generated by the identical source subject and by reducing the sample size to derive standard errors concomitantly.*"

We have also amended the results section (lines 176 to 207):

Epidemiologic and microbiologic risk factors of clonal and plasmid-mediated transmissions

To determine epidemiologic risk factors of clonal and plasmid-mediated transmissions, 1451 clonal transmission case pairs were compared with 1451 available control pairs, and 30,059 plasmid-mediated transmission case pairs with 30,059 available control pairs. Risk factors associated with clonal transmission in the multivariable analysis were direct ward contact (adjusted odds ratio [aOR], 6.22; 95%CI, 3.89–9.95; $P < 0.0001$), indirect ward contact (aOR, 2.90; 95%CI, 1.89–4.45; $P < 0.0001$), direct hospital contact (aOR, 4.66; 95%CI, 3.20–6.79; $P < 0.0001$) and indirect hospital contact (aOR, 1.62; 95%CI, 1.07–2.47; $P = 0.023$). Discipline and procedure contact were not associated with clonal transmission. Community contact was not analysed as only one of 1451 case pairs (0.07%) and no control pairs had any form of community contact (Table 1).

*Compared with *E. coli*, *K. pneumoniae* (aOR, 3.13; 95%CI, 2.14–4.58; $P < 0.0001$) and *Enterobacter* spp. (aOR, 2.29; 95%CI, 1.46–3.61; $P = 0.0003$) were positively associated with clonal transmission. Additionally, bacteria carrying *bla*NDM (aOR, 1.52; 95%CI, 1.04–2.22; $P = 0.031$) and *bla*OXA-type (aOR, 1.81; 95%CI, 1.13–2.91; $P = 0.014$) genotypes had increased odds of clonal transmission compared with *bla*KPC (Table 1).*

Risk factors demonstrating a weak association with plasmid-mediated transmission in the multivariable analysis were direct ward contact (adjusted odds ratio [aOR], 2.14; 95%CI, 0.63–7.27; $P = 0.22$), indirect ward contact (aOR, 1.46; 95%CI, 0.69–3.09; $P = 0.32$), direct hospital contact (aOR, 1.79; 95%CI, 0.98–3.27; $P = 0.058$) and indirect hospital contact (aOR, 1.20; 95%CI, 0.62–2.30; $P = 0.59$). However these contact risk factor associations did not meet criteria for statistical significance at the α level of 0.05. Discipline and procedure contact was not associated with plasmid-mediated transmission. Community contact was not analysed as only 11 of 30,059 case pairs

(0.04%) and two of 30,059 control pairs (0.007%) had any form of community contact (Table 2).”

We have also amended the Discussion section mainly in the following sections:
(lines 223 to 227):

“The independent association of indirect ward and indirect hospital contact with clonal CPE transmission suggest the presence of persistent reservoirs of CPE in the hospital environment. Our analysis found a weak association of contact risk factors with plasmid-mediated transmission which did not meet criteria for statistical significance.”

(lines 262 to 278):

“Plasmid-mediated transmission did not appreciably decrease during the study period which suggests limited impact of existing infection prevention measures. The weak association of contact risk factors (direct ward contact, indirect ward contact, direct hospital contact and indirect hospital contact) with plasmid-mediated transmission suggests that factors other than patient to patient contact may play a major role in carbapenemase plasmid dissemination. Prior studies have shown that conjugation of carbapenemase-encoding plasmids is associated with plasmid persistence in the hospital over prolonged periods, potentially as different species are suited to different ecologic niches[17]. Plasmid conjugation in varied bacterial species may facilitate transmission of the carbapenemase gene from an index patient to the inanimate environment, especially the aqueous environment, and subsequent transmission to another patient. Plasmid-mediated transmission and the contribution of hidden reservoirs to hospital CPE transmission is a domain which requires further study.”

The abstract has also been amended:
(lines 30 to 33)

“Indirect ward and hospital contact were identified as independent risk factors associated with clonal transmission (indirect ward adjusted odds-ratio [aOR], 2.90 [95%CI, 1.89–4.45; P<0.0001]; indirect hospital aOR, 1.62 [95%CI, 1.07–2.47; P=0.023].”

- General approach for data analysis: the control pairings are selected under certain conditions, but these may lead to a circular reasoning given the transmission locations, potentially. Could the authors please comment on that?

Reply:

We would not expect bias in the selection of controls to affect the association analysis with transmission locations as transmission location was not a criteria for selection of controls. Similarly the other epidemiologic contacts analysed, specifically community, discipline and procedure contact were not part of the selection criteria.

The selection of control pairings for clonal transmission was based on control acquisition subjects meeting the following criteria: (i) the control acquisition subject was not the same as the acquisition subject or source subject of the case pair; (ii) the date of culture of the isolate from the control acquisition subject was same as or later than the date of culture of the isolate from the acquisition subject in the case pair; and (iii) the isolates of the control acquisition subject did not meet clonal transmission criteria with the source subject or the acquisition subject in the case pair.

The selection of control pairings for plasmid-mediated transmission was based on control acquisition subjects meeting the following criteria: (i) the control acquisition subject was not the same as the acquisition subject or source subject of the case pair; (ii) the date of culture of the isolate from the control acquisition subject was same as or later than that from the acquisition subject in the case pair; (iii) the isolates of the control acquisition subject did not meet clonal or plasmid-mediated transmission criteria with the source subject or acquisition subject in the case pair.

- Please provide statistical programming code and if possible also the data for review.

Reply: We provide the programming code and data for review.

REVIEWERS' COMMENTS

Reviewer #4 (Remarks to the Author):

The authors have addressed all points I raised in the first review of the paper. The authors have added sensitivity analyses if certain assumptions could have affected the results. The statistical programming code was provided together with the data, as requested. In my comment 2.2 Data analysis, the authors described the adjustment (via weighting) of the standard errors due to clustering. An additional option for the estimation step would be to use so-called sandwich estimators to account for correlation within the data. I would leave this to the authors' decision as to whether they think this would alter the results. If they do not think so, there is no need for additional revision.

I downloaded the data and statistical programming code. It was running fine.

From my point of view the statistical methods are acceptable and the paper may be published, if the authors considered my suggestion regarding the additional adjustment of the standard errors.

REVIEWERS' COMMENTS

Reviewer #4 (Remarks to the Author):

The authors have addressed all points I raised in the first review of the paper. The authors have added sensitivity analyses if certain assumptions could have affected the results. The statistical programming code was provided together with the data, as requested. In my comment 2.2 Data analysis, the authors described the adjustment (via weighting) of the standard errors due to clustering. An additional option for the estimation step would be to use so-called sandwich estimators to account for correlation within the data. I would leave this to the authors' decision as to whether they think this would alter the results. If they do not think so, there is no need for additional revision.

I downloaded the data and statistical programming code. It was running fine.

From my point of view the statistical methods are acceptable and the paper may be published, if the authors considered my suggestion regarding the additional adjustment of the standard errors.

Reply: We thank the reviewer for the very helpful comments and do not think that the use of sandwich estimators would alter the results.